# Influenza-associated hospitalisation and mortality rates among global Indigenous populations; a systematic review and meta-analysis

Juliana M. Betts[1], Aaron L. Weinman[2], Jane Oliver[2], Maxwell Braddick[2,3],
Siyu Huang[2,4], Matthew Nguyen[2], Adrian Miller[5], Steven Y. C. Tong[2,3], Katherine
B. Gibney[2,3]*

**1** School of Public Health and Preventive Medicine, Monash University, Melbourne, Australia, **2** Department of Infectious Diseases, University of Melbourne at the Peter Doherty Institute for Infection and Immunity, Melbourne, Australia, **3** Victorian Infectious Disease Service, The Royal Melbourne Hospital at the Peter Doherty Institute for Infection and Immunity, Melbourne, Australia, **4** Melbourne Medical School, University of Melbourne, Melbourne, Australia, **5** Centre for Indigenous Health Equity Research, Central Queensland University, Townsville, Australia

* katherine.gibney@unimelb.edu.au

**Data Availability Statement:** All data underlying these findings is available in the original article and the supplementary material.

## Abstract

### Background

More than 50 million influenza infections and over 100,000 deaths from influenza occur annually. While Indigenous populations experience an inequitable influenza burden, the magnitude of this inequity has not previously been estimated on a global scale. This study compared rates of influenza-associated hospitalisation and mortality between Indigenous and non-Indigenous populations globally.

### Methods

A systematic review and meta-analysis was conducted including literature published prior to 13 July 2021. Eligible articles either reported a rate ratio (RR) comparing laboratory-confirmed influenza-associated hospitalisation and/or mortality between an Indigenous population and a corresponding benchmark population, or reported sufficient information for this to be calculated using publicly available data. Findings were reported by country/region and pooled by country and period (pandemic/seasonal) when multiple studies were available using a random-effects model. The $I^2$ statistic assessed variability between studies.

### Results

Thirty-six studies (moderate/high quality) were included; all from high or high-middle income countries. The pooled influenza-associated hospitalisation RR (HRR) for indigenous compared to benchmark populations was 5·7 (95% CI: 2·7–12·0) for Canada, 5·2 (2·9–9·3) for New Zealand, and 5.2 (4.2–6.4) for Australia. Of the Australian studies, the pooled HRR for seasonal influenza was 3.1 (2·7–3·5) and for pandemic influenza was 6·2 (5·1–7·5).

**Funding:** The authors received no specific funding for this work.

**Competing interests:** The authors have declared that no competing interests exist.

Heterogeneity was slightly higher among studies of pandemic influenza than seasonal influenza. The pooled mortality RR was 4.1 (3·0–5.7) in Australia and 3·3 (2.7–4.1) in the United States.

## Conclusions

Ethnic inequities in severe influenza persist and must be addressed by reducing disparities in the underlying determinants of health. Influenza surveillance systems worldwide should include Indigenous status to determine the extent of the disease burden among Indigenous populations. Ethnic inequities in pandemic influenza illustrate the need to prioritise Indigenous populations in pandemic response plans.

## Introduction

Influenza viruses and their pandemic potential remain a persistent threat to global health in the 21st century. Typically manifesting as fever and cough, influenza can rapidly progress to more severe illness resulting in hospitalisation and death, especially among children and the elderly [1]. The Global Burden of Disease Study estimated that in 2017 influenza caused 54.5 million lower respiratory tract infections; 8.2 million of which were severe and around 145,000 subsequent deaths occurred [2]. Indigenous populations, in particular, are known to be inequitably overrepresented in the influenza disease burden [3,4] although a systematic review demonstrating the extent of this inequity on a global scale is lacking. A greater understanding of those populations most at-risk of severe influenza is required to direct prevention and intervention strategies effectively, in keeping with the Public Health Research Agenda for Influenza developed by the World Health Organization (WHO) Global Influenza Program [5].

The term 'Indigenous' cannot be universally defined, due to the significant diversity present among the 370 million Indigenous people living in over 70 countries throughout the world today [6]. However, the term is generally interpreted to mean peoples who have a "historical continuity with pre-invasion and pre-colonial societies" [7], with communities having the right to self-identify as Indigenous "in accordance with their customs and traditions" [8].

In general, Indigenous populations throughout the world are known to experience higher rates of ill-health compared with non-Indigenous populations, although there is a lack of quality data from low and middle-income countries [9]. Anderson *et al.* demonstrated that while Indigenous groups from a range of countries typically experience shorter life expectancy, higher infant and maternal mortality rates, lower levels of education and higher rates of chronic disease when compared with corresponding benchmark populations, this is not uniformly the case, and the magnitude of the discrepancy differs based on the study setting [10].

Analyses of historical data from the Spanish influenza pandemic of 1918 indicate that Indigenous populations were disproportionately affected throughout the United States, Canada, Nordic countries and the Pacific, with estimated mortality rates ranging from 90% among Alaskan Inuits to 1·3% among native Hawaiians [4]. These rates are significantly higher than the estimated mortality rates for the corresponding non-Indigenous populations (ranging from 0·20% to 0·79%) [4].

More advanced diagnostic technologies such as reverse transcription polymerase chain reaction (PCR) testing has enabled a high level of diagnostic accuracy to be achieved when testing for influenza in modern times, as was the case during the 2009 H1N1 pandemic (2009pH1N1). As demonstrated in a multi-country comparison by La Ruche *et al.*,

2009pH1N1-associated hospitalisation and mortality among Indigenous populations was significantly higher when compared to benchmark populations in Canada (hospitalisation relative risk RR 5·7, mortality RR 3·4), the United States (hospitalisation RR 4·1, mortality RR 4·3), Brazil (hospitalisation RR 4·4), Australia (hospitalisation RR 7·7, mortality RR 5·1), New Zealand (hospitalisation RR 3·0) and New Caledonia (mortality RR 5·3) [3]. Almost one-hundred years following the Spanish influenza pandemic, the inequity in influenza disease burden with regard to Indigenous status remains stark.

It is therefore timely that a systematic review taking a global approach to the characterisation of influenza among Indigenous populations in both pandemic and inter-pandemic years be undertaken. The present study aimed to compare the rates of influenza-associated hospitalisation and mortality between Indigenous and non-Indigenous populations globally. This review has significant implications for informing vaccination policy, enhancing pandemic preparedness and addressing ethnic inequity.

## Methods

### Ethics statement

Formal ethics approval was not sought for this systematic review.

### Search strategy and selection criteria

A systematic review and meta-analysis of literature published prior to 13 July 2021 was undertaken following the guidelines outlined in the PRISMA statement [11]. The review was registered with the International Prospective Register of Systematic Reviews (PROSPERO) prior to data extraction (Registration No.: CRD42017075598). The following electronic databases were searched; Pubmed, Medline, Embase, Cochrane Central Register of Controlled Trials and CINAHL. Reference lists of studies were also reviewed for additional relevant papers.

Based on the nominated Indigenous groups outlined by Anderson *et al.*, [10] the following search terms were used (for full search strategy see **Table A in S1 Text**);

Indigenous OR Aborigin\* OR native OR trib\* OR First nation\* OR Maori OR Inuit\* OR Indians, North American OR (Torres Strait Island\*) OR Dai OR Tibet\* OR Mon OR Sherpa OR Rai OR Magar OR Tamang OR FATA OR Sami OR Nenet\* OR Baka OR Pygm\* OR Maasai OR Ijaw OR Fulani OR Metis OR Mapuche OR Kuna Yala OR (Embera Wounaan) OR Ngabe Bugle

AND

Influenza

AND

hospital\* OR mortality OR death OR fatal\*

Searches were undertaken by JB up to 13 June 2017 and were updated by JO from June 2017 to 13 July 2021. Selected data fields were abstracted and imported into the systematic review software Covidence [12]. Two independent reviewers assessed each record (JB, AW/ MN for searches conducted to 13th June 2017 and JO, CH for the updated searches) using a two-step screen. First, articles were screened by title and abstract, and second by full text review. Disagreements were arbitrated by a third independent reviewer to reduce errors (KG).

To be included, studies had to be published in English, and contain data from primary observational research. Included studies had an extractable parameter such that a rate ratio was reported or could be calculated comparing influenza-associated hospitalisation and/or mortality rates between the Indigenous population and the corresponding benchmark population of a particular country or region. We included only studies with laboratory-confirmed influenza, or those deemed most likely to have used laboratory confirmation based on the

study context. This was to ensure influenza alone was being studied, rather than other diseases with similar clinical presentations, such as pneumonia or other respiratory tract infections.

Single hospital-based studies that did not service states or regions with available denominator data, review articles, conference abstracts and studies which included cases based upon a clinical diagnosis of influenza alone were excluded. There were no timeframe or setting exclusions.

### Data analysis

Microsoft Excel was used to extract relevant data. This was performed by JB or JO and verified by another author (AW/MN).

For included studies, abstracted data fields (where available) were: country and setting, nominated Indigenous population and comparison benchmark population, study period, age group, vaccination status, comorbidities, influenza strain and whether seasonal or pandemic, Indigenous and benchmark population hospitalisation rates, hospitalisation rate ratio, Indigenous and benchmark population mortality rates, mortality rate ratio, 95% confidence intervals, case fatality rates and sample size. Not all studies reported all these data, including some that did not report 95% confidence intervals for rates and rate ratios.

Two reviewers (MB and SH) independently assessed individual study quality and risk of bias using the Joanna Briggs Institute (JBI) critical appraisal checklist for cohort studies. Discrepancies in assessment were adjudicated by a third reviewer (KG).

Hospitalisation and mortality rates and rate ratios were reported and/or calculated based upon the number of events divided by the population at risk and presented per 100,000 person-months to enable comparison between studies that covered varying time periods. The rate ratio of influenza-associated hospitalisation and death in Indigenous compared to non-Indigenous populations for each country/region was reported and/or calculated. A global pooled rate ratio was not estimated due to hypothesised data heterogeneity and because each country and Indigenous group is distinct, limiting the impact of such a measure. However, where there were multiple studies originating from one country, a pooled rate ratio was calculated overall, and by pandemic or seasonal strain, if the necessary numerator and denominator figures were available. If not specified in the study, denominators were obtained from relevant, publicly available census data, which may have altered the rate ratios from those published in the original study. Pooled estimates, 95% confidence intervals and forest plots were generated using the statistical software STATA 15.1 [13]. A random effects model was chosen based upon the significant differences between studies in terms of geographical location and population. 95% confidence intervals (95% CI) that did not include the value 1 were considered statistically significant. The $I^2$ estimate of heterogeneity was used to assess variability between studies.

### Role of the funding source

There was no funding source for this study.

## Results

**Fig 1** depicts numbers of articles screened, assessed for eligibility and included in the review (see **Table B in S1 Text** for a full list of excluded articles). Thirty-six studies were included for data extraction and are summarised in **Table C in S1 Text**. Of the included studies, 15 (42%) were from Australia [14–28], 10 (28%) were from the United States of America (USA) [29–38], six (17%) were from New Zealand [39–44], and four (11%) were from Canada [45–48]. Only one (3%) was from a high-middle income country (Brazil) [49], and there were no included studies from low-middle or low-income countries.

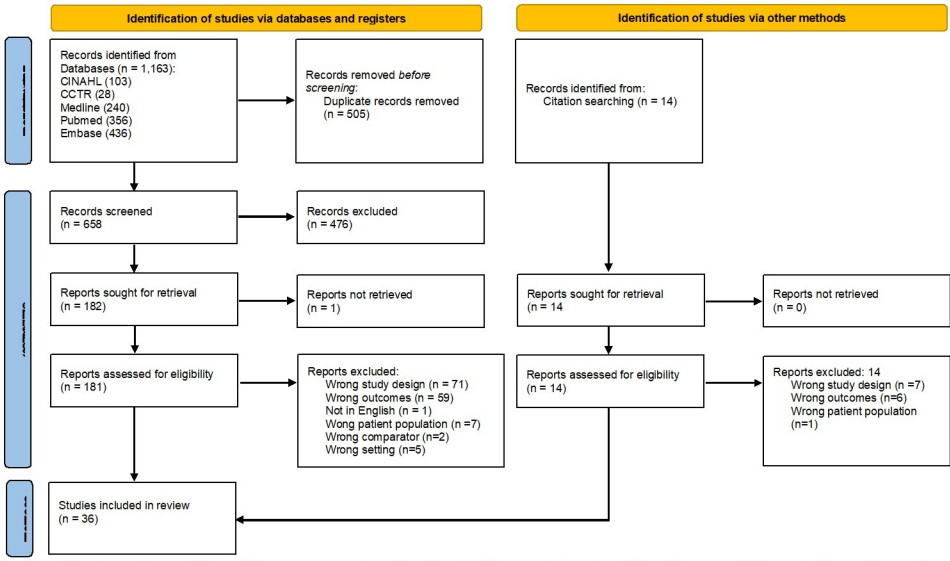

Note: The initial search (conducted on 13 June 2017) was updated on 13 July 2021. Figure 1 combines searches from June 2017 (n=875 records identified plus n=11 from citation search) and July 2021 (n=288 records records identified plus n=3 from secondary search).

From: Page MJ, McKenzie JE, Bossuyt PM, Boutron I, Hoffmann TC, Mulrow CD, et al. The PRISMA 2020 statement: an updated guideline for reporting systematic reviews. BMJ 2021;372:n71 doi: 10.1136/bmj.n71

**Fig 1. Flow diagram of included studies.**

Of the included studies, 21 (58%) had extractable data only for influenza-associated hospitalisation [14–18,22,23,31,33,34,38–44,46–49] and five (14%) had extractable data only for influenza-associated mortality [21,29,30,35,36]. There were 10 studies (28%) with extractable data for both hospitalisation and mortality [19,20,24–28,32,37,45]. Nearly three quarters of included studies (n = 26, 72%) analysed hospitalisations and/or deaths during the 2009pH1N1 [17–20,25–27,29–33,35–41,44–49], while seven (19%) analysed hospitalisations and/or deaths from seasonal influenza [14,15,22,23,34,42,43] and four (11%) covered both seasonal and pandemic periods [16,21,24,28]. Of the 36 included studies, 28 (78%) had no age limitations [17–32,34–40,44–46,48,49], while others considered children [14–16,33,43], adults [41] or women of reproductive age [42,47]. Data used to calculate hospitalisation and mortality rates is provided in **Table D in S1 Text.**

The JBI critical appraisal tool findings are summarised in **Table E and Table F in S1 Text.** Indigenous and non-Indigenous groups were assessed from the same population with the exception of one study, which compared the Indigenous population in Alaska with the non-Indigenous population in the rest of the USA, indicating a higher risk of bias [33]. There was heterogeneity in the methods for assessing Indigenous status. Studies using maternal identification and population projections to identify Indigenous cases, as opposed to patient identification, were classified as having an unclear risk of bias [14,16,21,25,26]. Studies using ICD codes rather than laboratory data to identify influenza infections were also classified as unclear in category 6 of the JBI tool [14,22–24,33–35]. Category 4 of the JBI tool (regarding identification of confounding factors) identified a low risk of bias for 30/36 studies. Category 5 of the JBI tool (regarding strategies to address confounding factors) identified 25/36 studies with a low risk of bias. Using the composite of the JBI critical appraisal tool, no studies assessed were classified as having an overall high risk of bias.

Influenza-associated hospitalisation rates among Indigenous populations ranged from 0.8 hospitalisations per 100,000 person-months among Native Americans and Alaska Natives in the United States during the first wave (Spring/Summer) of the 2009pH1N1 [32], to 89.7

hospitalisations per 100,000 person-months among Aboriginal and Torres Strait Islander peoples in Australia during the 2009pH1N1 (**Table 1**) [17].

A consistent pattern of statistically significant higher hospitalisation rates among Indigenous populations compared to benchmark populations was demonstrated (**Table 1**). The highest ethnic disparity was observed in Manitoba, Canada during the 2009pH1N1, whereby First Nations peoples were 16.1 (95% CI: 12·0–21·7) times more likely to be admitted to hospital for influenza compared with other Canadians [48]. The lowest hospitalisation rate ratio (HRR) was 1·2, observed in the United States over the seasonal influenza period 2001–2008 [34].

When observing pandemic influenza alone, HRRs comparing Indigenous to benchmark populations ranged from 1·4 in the United States [32], to 16·1 (95% CI: 12·0–21·7) in Canada [48]. When observing seasonal influenza alone, the highest disparity was observed among infants in Auckland, New Zealand, wherein Māori children were 11.1 (95% CI: 4.1–28.1) times more likely to be hospitalised for influenza compared with European and other children during the winter months of 2014–2016 [43]. The lowest seasonal influenza HRR was 1·2, observed in the USA over 2001–2008 [34].

For hospitalisations, the pooled overall influenza–associated HRR (Indigenous compared with the benchmark population) was 5·7 (95% CI: 2·7–12·0, n = 4 studies) for Canada, 5·2 (95% CI: 2.9–9.3, n = 3 studies) for New Zealand and 5.2 (95% CI: 4.2–6.4, n = 12 studies) for Australia (**Fig 2**).

Seven studies included data on seasonal influenza-associated hospitalisation (separate from pandemic influenza) [14,15,22,23,34,42,43], four (57%) of which were from Australia. For the three Australian studies with sufficient information, the pooled seasonal influenza HRR (Indigenous vs benchmark population) was 3.1 (95% CI: 2·7–3·5) (**Fig 3**). By contrast, for pandemic influenza, the pooled HRR in seven Australian studies was 6·2 (95% CI: 5·1–7·5). Heterogeneity was slightly higher among studies of pandemic influenza ($I^2$ = 93·7%) compared with seasonal influenza ($I^2$ = 79·1%).

Globally, the lowest rate of influenza-associated mortality for an Indigenous population was 0.03 deaths per 100,000 person-months, observed among Indigenous Australians in the Northern Territory over the period 2007–2016 (Table 2) [28]. The highest Indigenous mortality rate was 1.6 deaths per 100,000 person-months among Indigenous Australians in North Queensland during the 2009pH1N1, however this was based on only five deaths over the study period [19].

For benchmark populations, the lowest influenza-associated mortality rate was 0·0 deaths per 100,000 person-months, observed among benchmark populations of the United States [32] and Australia [24,28]. The highest mortality rate among benchmark populations was 0·5 deaths per 100,000 person-months, observed in North Queensland, Australia during the 2009pH1N1 [19] (**Table 2**).

Mortality rate ratios (MRR) tended to be greater than 1 (see **Table 2**), indicating an inequitable impact of influenza on Indigenous populations. The lower limit of the 95% CI was >1 for 12 of the 15 (80%) included studies [20,24–30,32,35,36,45]; crossed one for two studies [19,37]; and was not reported for one study [21]. The greatest disparity in mortality was observed in Australia, whereby Indigenous Australians were 5·9 (95%CI: 3.6–9.1; age-standardised 5·7) times more likely to die from 2009pH1N1 compared with non-Indigenous Australians [24]. The lowest MRR was also observed in Australia (MRR = 1·1) which was an average MRR for years 2006–2013 [21]. For this study, when the MRR for 2009 was excluded (MRR = 3·3), the average MRR decreased to 0·79 (95% CI not provided).

The pooled influenza-associated MRR (Indigenous vs. benchmark populations) was 4.1 (95% CI: 3·0–5.7, n = 7 studies) in Australia and 3.3 (95%CI: 2.7–4.1, n = 5 studies) in the United States (**Fig 4**).

**Table 1. Overall influenza-associated hospitalisation rates (HR) and hospitalisation rate ratios (HRR) (crude rates unless otherwise specified) for Indigenous peoples compared with a benchmark population.**

| Author, year | Country (region); age-group | Study Period | Indigenous HR (per 100,000 person-months) | Benchmark HR (per 100,000 person-months) | HRR | 95% Confidence Interval (HRR) | n |
|---|---|---|---|---|---|---|---|
| Baker et al, 2009 [39] | New Zealand (nation-wide); all ages | May—August 2009 | 10.8* | 3.5* | 3.0* | 2.9–3.2 | 972 |
| Bandaranayake et al, 2011 [40] | New-Zealand (nation-wide); all ages | January—October 2010 | - | - | 1.8 | 1.6–2.0 | 732 |
| Dee et al, 2010 [41] | New-Zealand (Hutt Hospital); adults >18 years | June—July 2009 | 81.3 | 15.7 | 5.2 | 2.7–9.8 | 54 |
| Prasad et al, 2019 [42] | New Zealand (Auckland); women of reproductive age | 2012–2015 | 16.9 | 4.7 | 3.6 | 2.5–5.1 | 123 |
| Prasad et al, 2020 [43] | New Zealand (Auckland); infants < 1 year | 2014–2016 (winter months) | 38.3** | 3.5** | 11.1** | 4.4–28.1 | 44 |
| Verrall et al, 2010 [44] | New Zealand (Wellington, Hutt Valley); all ages | June—August 2009 | 42.7* | 8.5* | 5.0* | - | 229 |
| Carville et al, 2007 [14] | Australia (Western Australia); age 0–2 years | 1990–2000 | 5.1 | 1.3 | 3.9 | 3.1–4.8 | 520 |
| D'Onise et al, 2008 [15] | Australia (South Australia); all ages | 1996–2006 | 13.5φ | 5.2φ | 2.6 | - | 649 |
| Fathima et al, 2018 [16] | Australia (Western Australia); age <16 years | 2000–2012 (Influenza A) | 2.7 | 0.4 | 7.2 | 5.5–9.4 | 228 |
| | | 2001–2012 (Influenza B) | 0.9 | 0.1 | 11.0 | 6.5–18.9 | 54 |
| | | 2000–2012 (Influenza A + B) | 2.1# | 0.3# | 7.9# | 6.2–10.1 | 282 |
| Flint et al, 2010 [17] | Australia (Northern Territory Top End); all ages | June—August 2009 | 89.7 | 9.7 | 9.3 | - | 161 |
| Goggin et al, 2011 [18] | Australia (Western Australia); all ages | June—August 2009 | 10.6† | 1.7† | 6.4† | 3.8–10.7 | 100 |
| Harris et al, 2010 [19] | Australia (North Queensland); all ages | May—August 2009 | 37.1 | 4.7 | 7.9 | 4.7–13.2 | 61 |
| Kelly et al, 2009 [20] | Australia (nation-wide); all ages | May—October 2009 | 25.1 | 3.2 | 7.8 | 7.2–8.4 | 4 833 |
| Menzies et al, 2004 [22] | Australia (nation-wide); all ages | July 1999—June 2002 | 4.1* | 1.4* | 2.9* (2.7) | 2.5–3.0 | 10 313 |
| Menzies et al, 2008 [23] | Australia (New South Wales, the Northern Territory, Queensland, South Australia and Western Australia); all ages | July 2002- June 2005 | 3.2* | 1.3* | 2.3* (3.0) | 2.7–3.2 | 7 378 |
| Naidu et al, 2013 [24] | Australia (New South Wales, Northern Territory, Queensland, South Australia, Victoria, Western Australia); all ages | July 2005- June 2010 | 8.1* | 1.8* | 4.6* | 4.4–4.9 | 22 998 |
| New South Wales Public Health Network, 2009 [25] | Australia (New South Wales); all ages | May—August 2009 | 17.3ʃ | 4.4ʃ | 4.0 | 3.2–4.9 | 1 214 |
| Pennington et al, 2018 [26] | Australia (nation-wide); all ages | January—December 2009 | 10.5 | 1.7 | 6.2 | 5.7–6.7 | 5 085 |
| Rudge et al, 2010 [27] | Australia (New South Wales); all ages | April—August 2009 | 7.8 | 1.9 | 4.2 (3.2*) | 3.5–5.3 | 1 131 |
| Weinman et al, 2020 [28] | Australia (Northern Territory); all ages | 2007–2016 | 2.0* | 0.3* | 6.5* | 5.9–7.2 | 2 107▽ |
| Chowell et al, 2012 [31] | U.S.A (Maricopa County); all ages | April 2009—April 2010 | - | - | 6.2 | 6.1–6.2 | 532 |

(*Continued*)

**Table 1.** (Continued)

| Author, year | Country (region); age-group | Study Period | Indigenous HR (per 100,000 person-months) | Benchmark HR (per 100,000 person-months) | HRR | 95% Confidence Interval (HRR) | n |
|---|---|---|---|---|---|---|---|
| Dee et al, 2011 [32] | U.S.A (10 states); all ages | April—August 2009 | 0.8* | 0.6* | 1.4* | - | 1 238 |
| | | September 2009—January 2010 | 6.5* | 3.3* | 2.0* | - | 4 637 |
| Foote et al, 2015 [33] | U.S.A (nation-wide); age < 1 year | April 2009—March 2010 | 82.5 | 27.5 | 3 | - | - |
| Gounder et al, 2014 [34] | U.S.A (IHS Contract Health Service Delivery Area counties (Indigenous) 13 states (benchmark)); all ages | 2001–2008 | 1.6φ | 1.3φ | 1.2 | - | - |
| Thompson et al, 2011 [37] | U.S.A (New Mexico); all ages | September 2009—January 2010 | 19.2* | 7.3* | 2.6* (2.2) | 1.8–2.6 | 926 |
| Wenger et al, 2011 [38] | U.S.A (Alaska); all ages | September—October 2009 | 28 | 7 | 4 | - | 96 |
| Helferty et al 2010 [45] | Canada (nation-wide); all ages | April 2009—April 2010 | 3.6† | 1.4† | 2.5† | 2.3–2.8 | 6 091 |
| Mostaço-Guidolin et al, 2013 [46] | Canada (Manitoba); all ages | May—August 2009 | 43.7† | 2.9† | 14.8† | 11.3–19.4 | 213 |
| | | October—January 2010 | 10.3† | 4.1† | 2.5† | 1.7–3.8 | 166 |
| Rolland-Harris et al, 2012 [47] | Canada (nation-wide, excluding Ontario and Nova Scotia); women of reproductive age | April 2009—April 2010 | 4.5ʃ | 1.1ʃ | 4.0ʃ | 3.4–4.8 | 986 |
| Zarychanski et al, 2010 [48] | Canada (Manitoba); all ages | April—September 2009 | 22.8†λ | 1.4†λ | 16.1†λ | 12.0–21.7 | 178 |
| Lenzi et al, 2012 [49] | Brazil (Paraná); all ages | January–December 2009 | 9.5β | 2.0β | 4.8β | 3.2–6.9 | 1 911 |

*age-standardised

† based on population data from the 2011 census [50,51]

β based on population data from the 2000 census [52]

λ based on population fraction of 0.072 [46]

ʃ based on population data from the 2006 census [53,54]

φ mean rate over study period

** based on population estimates provided [43];# based on cohort size of 469 589 children [16]

∇ figures provided through personal correspondence from authors; U.S.A–United States of America.

## Discussion

This review demonstrates that Indigenous populations from Australia, New Zealand, the United States, Canada and Brazil endure a disproportionate burden of severe influenza, in terms of hospitalisations and/or mortality, compared with corresponding benchmark populations. This was consistent across included studies, with only two studies having rate ratio confidence intervals that crossed one (both of which included small numbers of influenza deaths: n = 35 and 5 respectively [19,37]. These findings are consistent with those demonstrated by La Ruche *et al*., who indicated consistently higher hospitalisation and mortality rates for Indigenous populations throughout the Americas and the Pacific during the first wave of the 2009pH1N1 [3].

This disparity between Indigenous and non-Indigenous populations in influenza-associated hospitalisation and mortality demonstrates a significant health inequity. Health inequities are systematic, avoidable and unnecessary differences in health between groups of people which

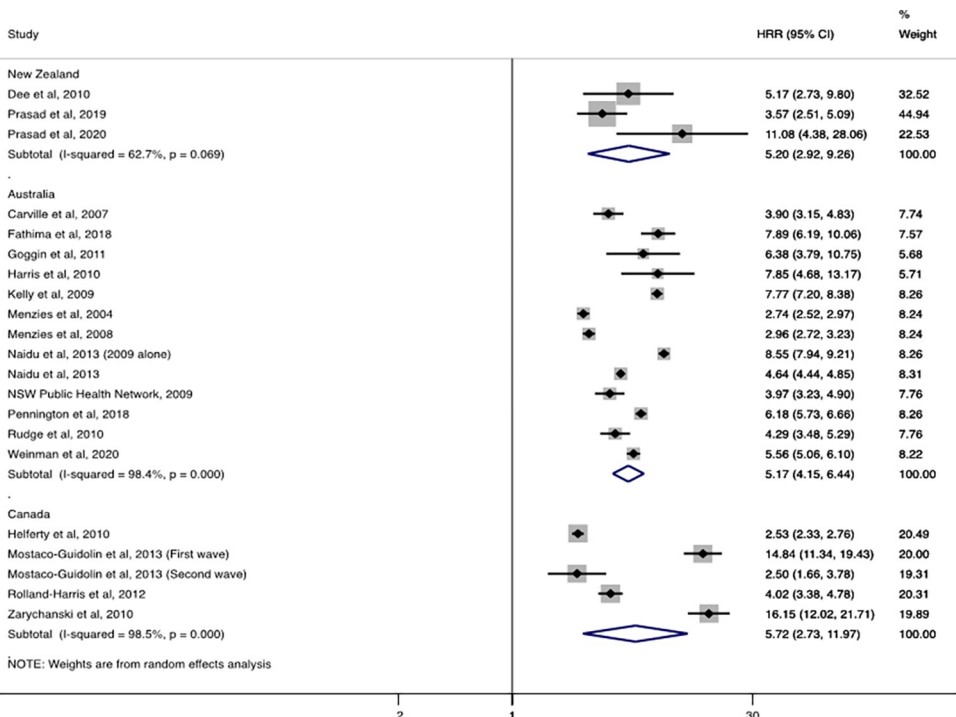

**Fig 2. Overall influenza-associated hospitalisation rate ratios (HRR) for Indigenous populations compared with a benchmark population by country.**

exacerbate underlying social disadvantage [56]. For many global Indigenous populations, including those from Australia, Canada, New Zealand, the United States and Brazil, the experience of colonialism is the common factor driving health inequities [58]. While particular circumstances differ between populations, the effects of violent dispossession from traditional

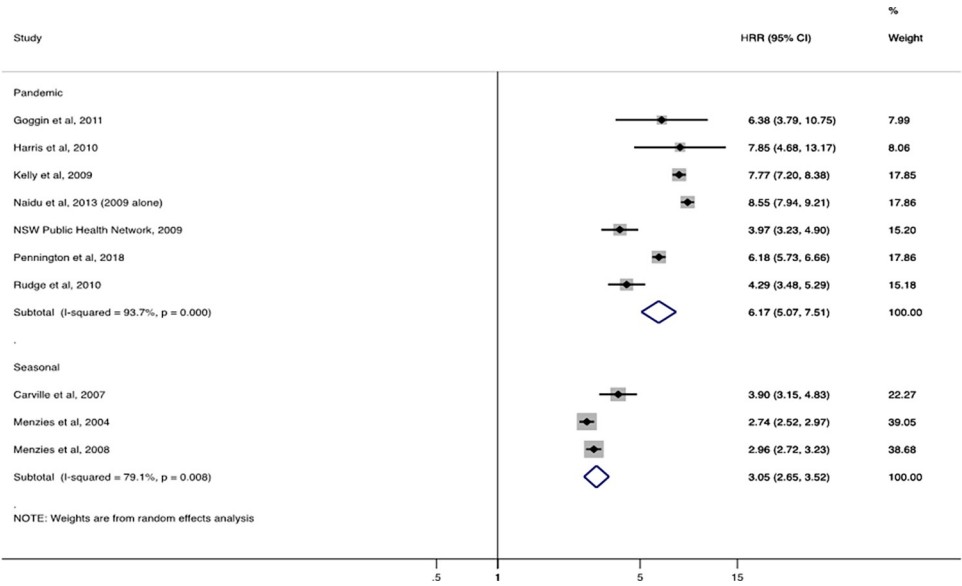

**Fig 3. Seasonal and pandemic influenza-associated hospitalisation rate ratios (HRR) from Australian studies for Indigenous populations compared with a benchmark population.**

**Table 2. Overall influenza-associated mortality rates (MR) and mortality rate ratios (MRR) (crude rates unless otherwise specified) for Indigenous peoples compared with a benchmark population.**

| Author, year | Country (region); age-group | Study Period | Indigenous MR per 100, 000 person-months | Benchmark MR per 100,000 person-months | MRR | 95% Confidence Interval | n |
|---|---|---|---|---|---|---|---|
| Brooks et al, 2012 [29] | U.S.A (New Mexico); all ages | January—December 2009 | 0.5 ^ | 0.2 ^ | 2.7 ^ | 1.4–5.3 | 50 |
| CDC, 2009 [30] | U.S.A (12 states **); all ages | April—October 2009 | 0.5* | 0.1 * | 3.5 (4.0*) | 2.5–4.7 (2.9–5.6*) | 426 |
| Dee et al, 2011 [32] | U.S.A (nation-wide); < 18 years of age | April 2009—March 2010 | 0.1 | 0.0 | 3 | 1.2–6.2 | 278 |
| Groom et al, 2014 [35] | USA (HIS contract health service delivery area counties); all ages | January—December 2009 | 0.4 | 0.1 | 4 | 3.8–6.4 | - |
| Hennessy et al, 2016 [36] | USA (Alaska, Arizona, New Mexico, Oklahoma, Wyoming); all ages | April—December 2009 | 0.4^ | 0.1^ | 3.8 ^ | 2.6–5.7 | 145 |
| Thompson et al, 2011 [37] | U.S.A (New Mexico); all ages | September 2009—January 2010 | 0.8^ | 0.4^ | 2.0 ^ | 0.8–4.8 | 35 |
| Harris et al, 2010 [19] | Australia (North Queensland); all ages | May—August 2009 | 1.6 | 0.5 | 3.2 | 0.4–29.0 | 5 |
| Kelly et al, 2009 [20] | Australia (nation-wide); all ages | May—October 2009 | 0.7 | 0.1 | 5.6 | 3.4–7.9 | 186 |
| Li-Kim-Moy et al, 2016 [21] | Australia (Western Australia, Northern Territory); all ages | 2010–2013 | - | - | 1.1 φ | - | 807 |
| Naidu et al, 2013 [24] | Australia (New South Wales, Northern Territory, Queensland, South Australia, Western Australia); all ages | 2006–2010 | 0.1ꟼ | 0.0ꟼ | 2.2ꟼ | 1.3–3.7 | 235 |
| New South Wales Public Health Network, 2009 [25] | Australia (New South Wales); all ages | May—August 2009 | 0.9ꟼ | 0.2ꟼ | 5.4ꟼ | 2.1–13.6 | 48 |
| Pennington et al, 2017 [26] | Australia (nation-wide); all ages | January–December 2009 | 0.3 | 0.1 | 4.6 | 3.0–7.0 | 188∇ |
| Rudge et al, 2010 [27] | Australia (New South Wales); all ages | April–August 2009 | 0.9ꟼ | 0.2ꟼ | 5.8ꟼ (4.5*) | 2.3–14.7 | 45 |
| Weinman et al, 2020 [28] | Australia (Northern Territory); all ages | 2007–2016 | <0.1* | <0.1* | 5.5* | 2.4–12.7 | 29 |
| Helferty et al 2010 [45] | Canada (nation-wide); all ages | April 2009 – April 2010 | 0.2 † | 0.1 † | 2.7† | 1.8–3.8 | 288 |

* age-standardised

^ based on population data from the 2010 census population [55]

† based on population data from the 2011 census [51]

ꟼ based on population data from the 2006 census [53]

φ mean over several years

** Alabama, Alaska, Arizona, Michigan, New Mexico, North Dakota, Oklahoma, Oregon, South Dakota, Utah, Washington, Wyoming; ∇ figures provided through personal correspondence from authors; U.S.A–United States of America.

lands, policies of exclusion and ongoing experiences of racism and discrimination, particularly within the healthcare sector, continue to unjustly manifest in worse health outcomes for Indigenous peoples [57–59].

Our findings from Australian studies demonstrate that the observed disparity between Indigenous and benchmark populations was more pronounced during 2009pH1N1 than with seasonal strains (HRRs 6·2; 95% CI: 5·1–7·5 and 3.1; 95% CI: 2·7–3·5 respectively). This is supported by the finding that the lowest rate ratios for both hospitalisation and mortality were observed among studies that included seasonal influenza periods [21,34]. The reason for this difference is not clear, however heterogeneity between the included studies was high, limiting

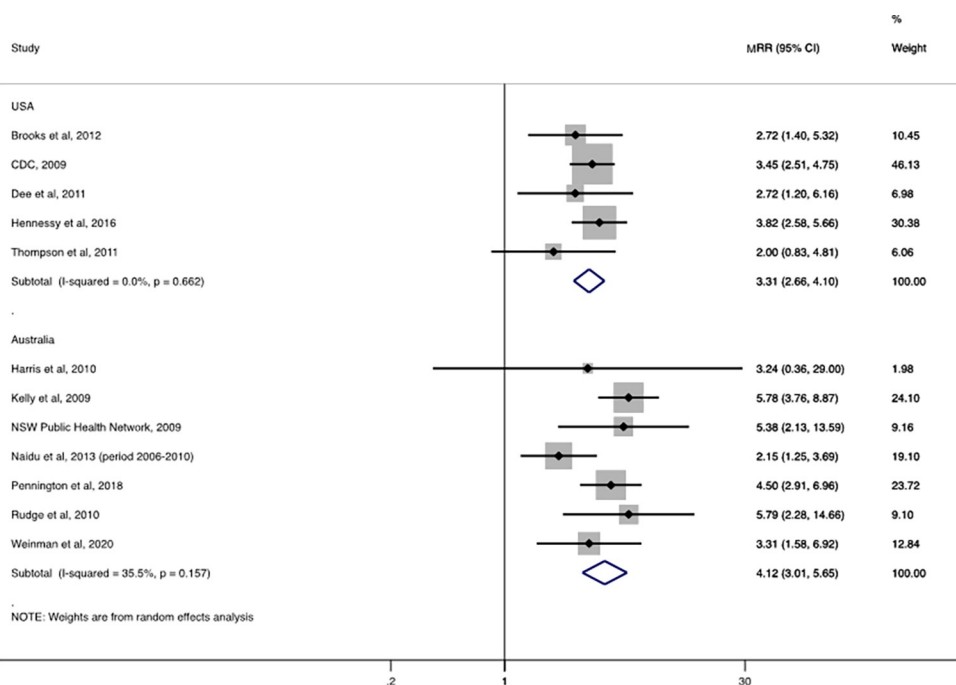

**Fig 4. Overall influenza-associated mortality rate ratios (MRR) by country for Indigenous peoples compared with a benchmark population.**

the reliability of the pooled estimates. One possible explanation is differing demographic profiles between Indigenous and benchmark populations. Indigenous populations in Australia, Canada, New Zealand and the United States are younger compared to corresponding benchmark populations [51,60–62] The 2009pH1N1 had a disproportionate effect on younger people, with the highest rates of notified disease among people aged 18–64 years, followed by children and adolescents aged 0–17 years [63]. In contrast, the majority of seasonal influenza hospitalisations and deaths occur among adults aged >65 years [1]. The age-standardisation of rates counteract this difference in population profile to an extent, although age-standardised rates were not consistently available.

Of note, data from the 2009pH1N1 dominated our findings, with only about one-third of included studies including seasonal influenza periods. Furthermore, the data on seasonal influenza was disproportionately found among Australian and New Zealand studies, with one study from the United States [34] and no studies from Canada or Brazil incorporating data on seasonal influenza. This reflects a limitation of our findings but also an important research gap in terms of sustaining influenza-related research during inter-pandemic periods to continue to understand priority populations for targeted public health interventions.

No eligible studies from low- and middle-income countries were identified, apart from one study from Brazil [49], an upper-middle income economy [64]. This, in part, reflects our inclusion criteria, which required publication in English, and laboratory-confirmation of influenza, as access to diagnostic technologies are limited in resource-poor settings. However, no eligible studies were identified from a number of other high-income countries with Indigenous populations (e.g. Norway, Denmark and Sweden) where legal prohibitions on the collection of ethnicity data exist [65]. The 2015 Sustainable Development Goals emphasised the need for more robust data collection which includes disaggregation by ethnicity, in order to achieve an accurate understanding of the health status of vulnerable populations to "leave no one behind" [66].

Hospitalisation and mortality rates were obtained from surveillance data, which is open to misclassification and outcome measurement bias. A number of included studies acknowledged that several regions within their country had imperfect reporting of ethnicity among cases. Similarly the data may have failed to account for secondary complications associated with influenza (for example bacterial pneumonia), whereby laboratory confirmation of the precipitating influenza infection may not have been sought. Consequently, the rates reported here likely underestimate the true extent of influenza-associated hospitalisation and mortality. Miscoding of data could have also led to missed cases of influenza, or accidental inclusion of other viruses with similar names (such as parainfluenza virus). Publication bias is also possible as studies that demonstrate a disparity between Indigenous and benchmark populations may be more likely to be published. However, for a number of the included studies, ethnic disparities were not the primary focus of the research.

The robust design of this review, in terms of its systematic approach and requirement for laboratory confirmation of influenza infection, is considered a strength. Included results are specific for influenza, rather than influenza and/or other respiratory illnesses. Previous studies, which fail to differentiate influenza from pneumonia, are generally more likely to document the effects of pneumonia as this typically accounts for 95% of undifferentiated pneumonia/ influenza cases [35]. Specifically identifying influenza is worthwhile because of its unique virological properties in relation to pandemic potential and vaccine-preventability.

The disproportionate burden of influenza-associated hospitalisation and mortality among Indigenous populations is a significant global health inequity which must be addressed. The results of this review have far-reaching public health implications for global Indigenous communities as well as policy-makers, health practitioners and researchers. Efforts to reduce underlying socioeconomic inequalities, enhance health-care access, ensure appropriate and timely use of antiviral medication, and improve the design and uptake of influenza vaccines are key strategies to be considered. Moreover, the development of surveillance systems which accurately capture both influenza and ethnicity, particularly for less well represented low- and middle-income countries, are necessary to characterise the extent of the influenza disease burden among Indigenous populations globally. The findings from this review also suggest a more severe disparity occurred during pandemic periods of influenza, which emphasises the worldwide need to prioritise Indigenous populations in influenza pandemic response plans. Future research should prioritise quantifying the influenza disease burden among Indigenous populations of low and middle-income countries, as well as improving the effectiveness of influenza vaccines among Indigenous populations.

## Research in context

### Evidence before this study

Numerous studies comparing the health status of Indigenous populations with benchmark populations have demonstrated poorer health outcomes generally as well as specifically for influenza. However, most studies are based upon clinical diagnoses of influenza and focus on pandemic, rather than seasonal periods. To our knowledge, no global systematic review has compared influenza-specific hospitalisation and mortality between Indigenous and benchmark populations during seasonal and pandemic periods.

### Added value of this study

This study provides a robust analysis of the available evidence relating to influenza hospitalisation and mortality for Indigenous populations compared with benchmark populations. Consistent across included studies, Indigenous populations from a number of countries endure

higher rates of both hospitalisation and mortality due to influenza and this disparity may be more pronounced for pandemic, rather than seasonal influenza. This systematic review also highlighted the void in research from low- and middle-income countries, where influenza surveillance systems and diagnostic capacity may be less developed.

### Implications of all the available evidence

Our findings have implications for influenza management strategies at both the population and individual level. Indigenous populations should be prioritised in influenza pandemic preparedness plans particularly with regard to the availability and uptake of influenza vaccines and antiviral medications. Policy-makers and health professionals should consider the wider contextual factors impacting upon the baseline level of health in Indigenous populations, and work together with Indigenous communities to achieve health equity. Future research should focus on characterising inequities between population groups in low- and middle-income countries as well as increasing the effectiveness of influenza vaccines among Indigenous populations.

## Supporting information

**S1 Checklist. PRISMA 2020 checklist.**
(DOCX)

**S1 Text.**

• Table A: Search Strategy

• Table B: List of excluded studies (from full text review)

• Table C: Features of included studies

• Table D: Notes about hospitalisation- and mortality-rate calculations

• Table E: Quality assessment of included studies

• Table F: JBI risk of bias assessment–comments.

• References.
   (DOC)

## Author Contributions

**Conceptualization:** Juliana M. Betts, Steven Y. C. Tong, Katherine B. Gibney.

**Data curation:** Juliana M. Betts, Aaron L. Weinman, Siyu Huang, Matthew Nguyen, Katherine B. Gibney.

**Formal analysis:** Juliana M. Betts, Jane Oliver, Maxwell Braddick, Siyu Huang, Steven Y. C. Tong, Katherine B. Gibney.

**Investigation:** Juliana M. Betts.

**Methodology:** Juliana M. Betts, Steven Y. C. Tong, Katherine B. Gibney.

**Supervision:** Steven Y. C. Tong, Katherine B. Gibney.

**Validation:** Aaron L. Weinman.

**Visualization:** Juliana M. Betts, Maxwell Braddick.

**Writing – original draft:** Juliana M. Betts.

**Writing – review & editing:** Juliana M. Betts, Aaron L. Weinman, Jane Oliver, Maxwell Braddick, Siyu Huang, Matthew Nguyen, Adrian Miller, Steven Y. C. Tong, Katherine B. Gibney.

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
