## [Editor Report · Decision Letter 0]

15 Nov 2022

PGPH-D-22-01700

Influenza-associated hospitalisation and mortality rates among global Indigenous populations; a systematic review and meta-analysis

Dear Dr. Betts,

Thank you for submitting your manuscript to PLOS Global Public Health. After careful consideration, we feel that it has merit but does not fully meet PLOS Global Public Health’s publication criteria as it currently stands. Therefore, we invite you to submit a revised version of the manuscript that addresses the points raised during the review process.

We look forward to receiving your revised manuscript.

Kind regards,

Javier H Eslava-Schmalbach, M.D., Ph.D., MSc

Academic Editor

Journal Requirements:

2. We have noticed that you have uploaded Supporting Information files, but you have not included a list of legends. Please add a full list of legends for your Supporting Information files after the references list. 

Additional Editor Comments (if provided):

Dear Authors:

Thanks for submitting your article to our journal. To facilitate the process of our reviewers and future readers, please add to the supplementary material: the searching strategy and the table of excluded articules (with the reason of exclusion). In the line 176, double-check if it is right the text "second reviewer". Also, double-check the title in the Table 2 (supplementary file).

Given that you submitted by first time this review on February 2022, I will consider, after your adjustments, to send your article to our reviewers. However, if they comment that the paper should be updated (to be less than one year old), I will accept their suggestions. I mention this, to prepare you, in case this happen.

Additionally, I suggest to draw the PRISMA figure using the classic format, consolidating the final updated strategy, with the previous one. Within the boxes, or at the foot of the Figure, you could include the respective explanations.

Please send us these initial adjustments, to be able to submit it to the reviewers.
---

## [Decision Letter · Decision Letter 1]

17 Feb 2023

PGPH-D-22-01700R1

Influenza-associated hospitalisation and mortality rates among global Indigenous populations; a systematic review and meta-analysis

Dear Dr. Betts,

Thank you for submitting your manuscript to PLOS Global Public Health. After careful consideration, we feel that it has merit but does not fully meet PLOS Global Public Health’s publication criteria as it currently stands. Therefore, we invite you to submit a revised version of the manuscript that addresses the points raised during the review process.

We look forward to receiving your revised manuscript.

Kind regards,

Javier H Eslava-Schmalbach, M.D., Ph.D., MSc

Academic Editor

Journal Requirements:

2. We have noticed that you have uploaded Supporting Information files, but you have not included a list of legends. Please add a full list of legends for your Supporting Information files after the references list.

Additional Editor Comments (if provided):

Dear authors: we have received reviewers' comments. Please comment/answer/include all of their comments and suggestions.

Reviewers' comments:

Reviewer's Responses to Questions

**Comments to the Author**

1. If the authors have adequately addressed your comments raised in a previous round of review and you feel that this manuscript is now acceptable for publication, you may indicate that here to bypass the “Comments to the Author” section, enter your conflict of interest statement in the “Confidential to Editor” section, and submit your "Accept" recommendation.

Reviewer #1: (No Response)

Reviewer #2: All comments have been addressed

2. Does this manuscript meet PLOS Global Public Health’s publication criteria? Is the manuscript technically sound, and do the data support the conclusions? The manuscript must describe methodologically and ethically rigorous research with conclusions that are appropriately drawn based on the data presented.

Reviewer #1: Yes

Reviewer #2: Yes

3. Has the statistical analysis been performed appropriately and rigorously?

Reviewer #1: Yes

Reviewer #2: Yes

4. Have the authors made all data underlying the findings in their manuscript fully available (please refer to the Data Availability Statement at the start of the manuscript PDF file)?

Reviewer #1: Yes

Reviewer #2: (No Response)

5. Is the manuscript presented in an intelligible fashion and written in standard English?

Reviewer #1: Yes

Reviewer #2: Yes

6. Review Comments to the Author

Reviewer #1: The article raises a very important issue of health equity pertaining to the indigenous population. the manuscript however requires English editing to ensure usage of consistent and standard terminology throughout. Eg- "2009pH1N1" being used to denote the 2009 influenza pandemic. The manuscript would have been easier to review if the tables and figures were kept at the end of the main body instead of in between the body of the text.

Reviewer #2: This is a properly executed systematic review and meta-analysis on an important topic. I only have one comment (I am reviewing a revised version of the manuscript): country-level estimates for both USA and Canada rely solely on data from the 2009-10 H1N1 pandemic. This not only represents a limitation of the data, but also an important knowledge gap that needs to be addressed in future work. I think that this point warrants a sentence or two in the Discussion

7. PLOS authors have the option to publish the peer review history of their article (what does this mean?). If published, this will include your full peer review and any attached files.

**Do you want your identity to be public for this peer review?** For information about this choice, including consent withdrawal, please see our Privacy Policy.

Reviewer #1: **Yes: **Siddhartha Saha

Reviewer #2: **Yes: **Jesse Papenburg

---

## [Editor Report · Decision Letter 2]

8 Mar 2023

Influenza-associated hospitalisation and mortality rates among global Indigenous populations; a systematic review and meta-analysis

PGPH-D-22-01700R2

Dear Dr Betts,

We are pleased to inform you that your manuscript 'Influenza-associated hospitalisation and mortality rates among global Indigenous populations; a systematic review and meta-analysis' has been provisionally accepted for publication in PLOS Global Public Health.

Best regards,

Javier H Eslava-Schmalbach, M.D., Ph.D., MSc

Academic Editor